# On the Interplay Between Sparsity and Training in Deep Reinforcement Learning

## Abstract

We study the benefits of different sparse architectures for deep reinforcement learning. In particular, we focus on image-based domains where spatially-biased structures are common, such as those provided by convolutional nets. Using these and several other architectures of equal capacity, we show that sparse structure has a significant effect on learning performance. We also observe that choosing the best sparse architecture for a given domain depends on whether the hidden layer weights are fixed or learned.

## 1 Introduction

A fundamental principle of deep learning is that neural networks with many connections can represent more functions than sparse networks with fewer connections. However, this increased representational capacity comes at the cost of more computational resources, both in terms of storage (for network weights) and time (for running a forward pass). While computationally efficient, sparse networks also provide statistical efficiency when their connection assignments (i.e. *sparse structure*) accord with the dependencies expressed in the data distribution (i.e. *dependence structure*). This is why convolutional networks are thought to generalize well in settings with natural, spatial imagery (Krizhevsky et al., 2012; Simonyan & Zisserman, 2014; LeCun et al., 2015; Szegedy et al., 2015).

In deep reinforcement learning, prior knowledge of the environment guides which connections to represent. For instance, convolutional networks are common in domains with spatial dependence structure; these networks exploit the co-variability of nearby pixels by connecting neighboring inputs. Of course an environment's dependence structure is not always known, and in such cases one is assumed, or a fully-connected network is used. Efficiency suffers in the latter case as additional updates are needed to shrink the weights of weakly-related inputs. While these approaches underpin many performant learning systems (Mnih et al., 2015; Silver et al., 2017b; Hessel et al., 2018), they are both far from ideal in the general setting.

Methods to learn sparse structure have been studied. Graesser et al. (2022) recently benchmarked various sparse-training techniques in reinforcement learning (RL). Consistent with previous studies, they found that an agent's performance depends on sparse structure when the full network is learned end-to-end. Other work from RL reaches a similar result using fixed networks with random weights (Gaier & Ha, 2019; Martin & Modayil, 2021). These findings collectively motivate an investigation of the interplay between sparse structure and the learning strategy. A key question is whether the effects of sparsity are consistent across networks with weights fixed to random values and networks with learned weights.

Our paper presents two main findings. First, we confirm that **when controlling for network capacity, the choice of sparse structure has a significant effect on an agent's performance.** This result establishes previously unknown details about the deep RL domain of MinAtar (Young & Tian, 2019), while corroborating the findings from Graesser et al. (2022) and Martin & Modayil (2021). Second, we demonstrate that **the ordering in performance among sparse structures depends on whether the hidden layer weights are fixed or learned.** We examine whether sparse architectures—also known as sparse networks—that excel with a fixed hidden layer remain dominant when fully learned. In other words, we attempt to decouple the relative performance gains due to the network's sparse topology from those due to backpropagation; this is important to help us understand what makes a sparse structure beneficial. Somewhat surprisingly, we

observe that spatial structure—which convolutional networks exploit—is not always the most performant in domains with apparent spatial dependence. Additionally, we found that some architectures outperform their contenders when used with random weights compared to learned weights.

## 2 Problem Setting

This work studies deep reinforcement learning problems. The setting is characterized by an agent and environment that interact through an interface of actions $\mathcal{A}$ and image-based observations $\mathcal{O}$. At every time-step $t \in \mathbb{N}$, the agent chooses an action $a_t \in \mathcal{A}$ after receiving the observation $o_t \in \mathcal{O}$. It takes the action then receives the next observation $o_{t+1}$ along with a scalar reward $r_{t+1}$. A history of interaction is denoted $h = a_1 o_1, a_2 o_2, \cdots$, with length-$n$ histories coming from the set $\mathcal{H}_n \equiv (\mathcal{A} \times \mathcal{O})^n$, and all finite-length histories from $\mathcal{H} \equiv \cup_{n=1}^{\infty} \mathcal{H}_n$. Furthermore, the agent is assumed to observe samples from a distribution $e \colon \mathcal{H} \times \mathcal{A} \to \Delta(\mathcal{O} \times \mathbb{R})$, which conditions on the current history and action; this is the environment. In our setting, it is assumed that the environment allows histories to be repeated in an episodic fashion.

The goal is to learn a policy, $\pi \colon \mathcal{H} \to \Delta(\mathcal{A})$ that maximizes the expected sum of future discounted rewards. For a given discount factor $\gamma \in [0, 1)$, the action-value, $q_\pi(h, a)$, expresses the utility of taking action $a$ from the history $h$ and following $\pi$ for all time-steps thereafter:

$$q_\pi(h, a) = \mathbb{E}_{\pi, e}[R_{t+1} + \gamma R_{t+2} + \gamma^2 R_{t+3} + \cdots | H_t = h, A_t = a]. \tag{1}$$

In many deep RL settings, it is common for the agent to follow an $\epsilon$-greedy policy; this selects uniform-random actions with probability $\epsilon$ and otherwise selects actions that maximize the current action-value.

The full history requires an unbounded amount of memory to represent, Therefore, the agent maintains a finite internal state $s \in \mathcal{S}$. Following prior work (Dong et al., 2022; Sutton, 2022; Abel et al., 2023), we define the internal state recursively, as $s_{t+1} \equiv f(s_t, a_t, o_{t+1})$, for all time-steps $t$, with $f \colon \mathcal{S} \times \mathcal{A} \times \mathcal{O} \to \mathcal{S}$ as the state-update function. At any moment, $s_t$ is assumed to provide sufficient context for the agent's present circumstances in the environment, i.e. $s_t = h_t$. The terms "state" and "internal state" are henceforth used interchangeably.

### 2.1 RL with a Deep Q-Network

The Deep Q-Network (DQN) is a method for learning an approximate action-value function represented as a deep neural network (Mnih et al., 2015). The Deep Q-Network (DQN) is a method for learning an approximate action-value function represented as a deep neural network (Mnih et al., 2015). Its state-update function $\hat{f} \colon \mathbb{R}^d \to \mathcal{S}$ uses a common set of hidden-layer weights $\Phi$ to map input images, represented as $d$-dimensional vectors, to internal states: $f(s_t, a_t, o_{t+1}) \equiv \hat{f}(o_{t+1}; \Phi)$. In our work, we do not assume partial observability; the agent's internal state $\hat{f}$ is a neural network with a single hidden layer and ReLU activation functions. Furthermore, action-values are computed with a linear combination of the state and final-layer weights $w_a$, for each $a \in \mathcal{A}$:

$$\hat{q}(o, a; \theta) = w_a^\top \hat{f}(o; \Phi). \tag{2}$$

The full collection of parameters is denoted $\theta \equiv \{\Phi, w_a \, \forall a \in \mathcal{A}\}$. Network parameters are learned by minimizing the following loss, averaged over a minibatch $\mathcal{D}$ of experience:

$$L(\theta) = \frac{1}{|\mathcal{D}|} \sum_{(o_t, a_t, r_{t+1}, o_{t+1}) \in \mathcal{D}} \left[ (r_{t+1} + \gamma \arg\max_{a' \in \mathcal{A}} \hat{q}(o_{t+1}, a'; \bar{\theta}) - \hat{q}(o_t, a_t; \theta))^2 \right]. \tag{3}$$

The target term, $\hat{q}(o_{t+1}, a_{t+1}; \bar{\theta})$, is computed with a separate network of identical architecture but different parameters $\bar{\theta}$. This design prevents gradients from affecting the update and promotes optimization stability (Asadi et al., 2022). Every few cycles, $\bar{\theta}$ is assigned the current values of $\theta$. In our study, we use Adam (Kingma & Ba, 2014) to optimize network parameters.

## 2.2 Sparse Deep Q-Networks

Using DQN, our study examines the performance benefits of different sparse structures. We encode sparse structure as a binary matrix $M \in \{0, 1\}^{d \times n}$ and apply it by taking an element-wise product with the hidden-layer weights $\Phi \in \mathbb{R}^{d \times n}$, denoted $M \odot \Phi$. Preactivations are computed by taking the dot-product of the sparse weight matrix and an observation $o$. For a single-layer architecture, a forward pass is given by $\hat{q}(o, a) \equiv w_a^\top \hat{f}\big((M \odot \Phi)^\top o\big)$. We assume that the sparse structure is fixed throughout learning, so $M$ is not affected by optimization.

There are different ways to arrive at the sparse configuration of the binary masks $M$: some can be hand-crafted based on the designer's domain knowledge, while others can be learned. For instance, one of our baselines relies on $L_1$-regularization to induce sparsity in an end-to-end manner (Hastie et al., 2009; Hernandez-Garcia & Sutton, 2019; Ma et al., 2019; De & Doostan, 2022). In machine learning, $L_1$-regularization is a commonly used technique in which we add a term to the loss function and weight it by a regularization coefficient $\beta \in [0, 1)$:

$$L(\theta) = L(\theta) + \beta ||\theta||_1 \tag{4}$$

As $\beta$ approaches 1, the loss penalizes weights that are non-zero. Although the weights are not guaranteed to reach zero exactly due to finite steps of optimization and floating point approximation, at the end of training we can take sufficiently small weights in $\Phi$ and zero out their corresponding entries in the mask $M$. The rationale is that the smallest weights in $\Phi$ likely correspond to those that contribute less in generating useful features.

## 3 Related Work

**Sparse Networks in Supervised Learning.** Sparse networks are studied extensively in the context of supervised learning. For instance, Zimmermann et al., (2012) identify that sparse weight matrices in recurrent nets can prevent arithmetic overflows in high-dimensional time-series problems (Zimmermann et al., 2012). More recently, Frankle & Carbin (2018) demonstrate the existence of sparse networks that perform as well as fully-connected networks. Others study methods for obtaining such lottery tickets, as they are known, by pruning connections with small weights and regrowing connections with large gradients (Evci et al., 2020). Sokar et al. (2022) introduce the Dynamic Sparse Training (DST) method, which periodically prunes connections with small weight magnitudes and randomly adds new connections during training. In this setting, network sparsity has also been learned end-to-end by including connection masks as a learnable parameter (Liu et al., 2020). In the time-series classification problem, Xiao et al. (2022) apply DST to the kernels of a convolutional neural network. A later work studied the performance benefits of various pruning techniques on other challenging classification tasks (Xiao et al., 2024); surprisingly, they found that sparse networks can sometimes surpass their dense counterparts. In an extensive literature review, Hoefler et al., described works that maintained the network sparsity fixed while learning the weights (Hoefler et al., 2021). However, none of these works studied the different extents that learning can improve performance across a range of fixed sparse topologies.

**Sparse Networks in Reinforcement Learning.** Sparse networks have received comparatively less attention in RL. A line of work has applied DST with various pruning heuristics (Sokar et al., 2022; Graesser et al., 2022; Tan et al., 2022; Grooten et al., 2023; Obando-Ceron et al., 2024). Graesser et al., conducted an extensive empirical study that surveyed various techniques for adapting the network connectivity of DQN (Graesser et al., 2022); they established the conditions when sparse networks perform best in Atari and MuJoCo. Ceron et al., applied gradual pruning where the percentage of sparsity was smoothly adjusted during the course of training (Obando-Ceron et al., 2024). In multi-task RL, recent work used a gradient-based saliency criterion to select a subset of weights in the network—called a neural pathway—for each task (Arnob et al., 2025); this helped prevent catastrophic interference. All these works studied the effects of sparse structure while the full network was learned end-to-end. In another line of work, sparse structure was studied using fixed networks of random weights (Gaier & Ha, 2019). Martin & Modayil (2021) adapt the network structure based on predictions of the input observations. Modayil & Abbas (2023) extended this

technique to large-scale control settings. However, these works do not explore the performance benefits of sparsity when controlling for the learning process—fixing versus learning the hidden layer weights.

**Connection between Representations and the Environment.** Previous studies dealt with the problem of learning representations with no a priori knowledge of the environment's dependence structure. For instance, there are existing techniques for selecting subsets of sensor readings based on correlations in order to form a low dimensional embedding (Modayil, 2010). More importantly, this work showed that we can recover the underlying distribution of the raw inputs from the embedding, implying that the latter encoded informative features of the raw data. However, this technique was not applied in the online RL setting.

Moreover, other works have shown that one might not always need to learn the weights of a network approximator in order to represent a function of interest. For instance, in the supervised learning setting, Rahimi and Recht analytically showed that a shallow network with random hidden layer features and learned outer layer weights can generate a classifier that is not much worse than one where we optimally tune the non-linearities (Rahimi & Recht, 2008). In an earlier work, the authors also showed results suggesting that in some regression and classification tasks, simple linear approximators applied to random features of the inputs can outperform kernel machines (Rahimi & Recht, 2007). Furthermore, much earlier, Sutton and Whitehead suggested that random linear projections can result in useful and complex features in the online learning setting (Sutton & Whitehead, 1993).

Putting these two ideas together—that sparse subsets of the inputs can lead to informative features and that random linear projections of the inputs can be useful—an algorithm called Prediction Adapted Networks uses long-term predictions of the raw inputs to adapt the sparse connectivity of a value network with random hidden layer weights (Martin & Modayil, 2021; Modayil et al., 2014). Although this technique operates fully online and incrementally, its study is restricted to the RL prediction problem, where the learning objective is to estimate the sum of discounted rewards, conditioned on a fixed policy. As observed by the authors, predictions of the raw inputs can be used to uncover the underlying inter-dependencies in the data distribution within their chosen domain. To the best of our knowledge, no analysis of the relative performance of predictive sparsity when the hidden layer weights are random versus learned has been done in RL control.

## 4 Empirical Study

We show supporting evidence for the claims in Section 1, namely that (1) sparse structure has a significant effect on an agent's performance when controlling for network capacity, and (2) the ordering in performance among sparse structures depends on whether the hidden layer weights are fixed or learned. Comparisons are made measuring the return across time-steps, averaged over thirty independent trials after sweeping over the step-sizes. For complete details of our methodology, please refer to the Appendix.

### 4.1 The MinAtar Environment

MinAtar is a suite of simplified Atari 2600 video games (Young & Tian, 2019). Similar to the Arcade Learning Environment (Bellemare et al., 2013), MinAtar provides image observations, joystick commands, and game-score rewards. The games most relevant to our study of sparse structure are *Breakout* and *Space-Invaders*, as their object dynamics appear to have sparse, spatial relationships.

*Breakout* requires an agent to control a paddle and deflect a moving ball into a wall of bricks. A brick is destroyed whenever the ball contacts it, and the goal is to destroy as many bricks as possible. If the ball moves past the paddle, then the game is reset, and the score goes to zero. Observations are images with four channels, showing pixels of the paddle, ball, previous ball location, and bricks.

On the other hand, *Space-Invaders* is comparatively more complex. Here the agent commands a spaceship: controlling its horizontal position and cannon. The cannon fires munitions upward in straight lines. Above the ship is a row of aliens who move side to side and fire their cannons downward. If the ship is hit, the game resets and the score goes to zero. The goal is to shoot as many aliens as possible while avoiding their

attacks. Image observations have six channels that display locations of the ship, aliens, and munitions. For more information about both games, see the Appendix and the paper by Young & Tian (2019).

For our purposes, these games provide experiential data with spatially-dependent observations. In these domains, we expect spatially-biased architectures, whose topology hard-codes nearest-neighbor relationships, to perform best. In *Breakout*, for instance, the previous position of the ball is predictive of where it will be at the next time step. Thus, one expects connections from nearby pixels to be relevant, and pixels from distal parts of the image to be extraneous.

## 4.2 Architecture Baselines

We consider several sparse baseline architectures while controlling for the amount of sparsity and network capacity. Each architecture has the same number of learnable parameters and hidden layer dimensionality. In all the experiments, the degree of sparsity was fixed to 91% relative to a fully-connected architecture. This was controlled by setting the same number of zeroes in each architecture's binary mask. We chose 91% as we wanted the number of hidden layer connections to match the number of active inputs in a convolutional layer's $3 \times 3$ kernel used in the original MinAtar paper. See the Appendix for further details.

The first architecture, Random, imposes sparse connections in the hidden layer uniformly at random. Outperforming Random with another sparse architecture suggests that the underlying data distribution imposes non-random, sparse relationships among the observation components.

We also consider a spatially-biased baseline, Spatial. This architecture is similar to a convolutional layer in how it forms receptive fields with nearby pixels. However, to control for the effects of sparse structure and control for the number of learnable parameters, Spatial does not impose weight sharing nor equivariance to translations; each kernel contains its own learnable weights. This is the reason why our experiments do not include convolutional nets. For instance, if the ranking of a convolutional net became higher or lower compared to the other networks when the weights are learned, we would not know which of its properties (spatially-biased sparse structure, weight sharing or equivariance) accounts for this.

A third architecture, Predictive, establishes connections with the Prediction Adapted Networks algorithm (Martin & Modayil, 2021), which uses a measure of temporal relevance to assign its sparse connections. This baseline represents an architecture with non-random and non-spatial structure. Previous work has established its relevance in RL-based domains (Martin & Modayil, 2021; Modayil & Abbas, 2023).

Another baseline we consider, $L_1$-Reg, uses $L_1$-regularization to induce a sparse structure. It serves as an example for how an end-to-end algorithm can be used to obtain a sparse hidden layer structure. We controlled for the amount of sparsity in $L_1$-Reg by sweeping over the regularization coefficient until it matched the other architectures. Specifically, a weight is zeroed out if its final value is smaller than the average.

The final architecture we consider is fully-connected (Dense), meaning that each input influences all features in the hidden layer. No binary matrix is imposed onto the hidden layer weight matrix; thus this architecture has nearly ten times the number of active hidden layer weights as the sparse architectures.

Random, Spatial, and Predictive generate each pre-activation from the same number of inputs: 36 inputs (out of 400) generate each feature in *Breakout*, and 54 inputs (out of 600) in *Space-Invaders*. In contrast, the number of inputs used per feature by the $L_1$ baseline can vary, since it tries to maximize sparsity in aggregate, over the entire network. The Appendix provides more architectural details as well as visualizations for each type of sparse neighborhood in each environment.

## 4.3 Case Study: Fixed Hidden Weights and Topology

This experiment reveals that network sparsity plays an important role in the agent's performance when the hidden layer weights are randomly initialized and then held fixed. Moreover, these results also show that no architecture is strictly better across both environments in this setting. These are surprising results because, in practice, a single architecture is often thought to apply equally well across an entire suite of games, like MinAtar, ALE (Bellemare et al., 2013), Go, chess, shogi (Silver et al., 2017a; Schrittwieser et al., 2020), and Gran-Turismo (Wurman et al., 2022).

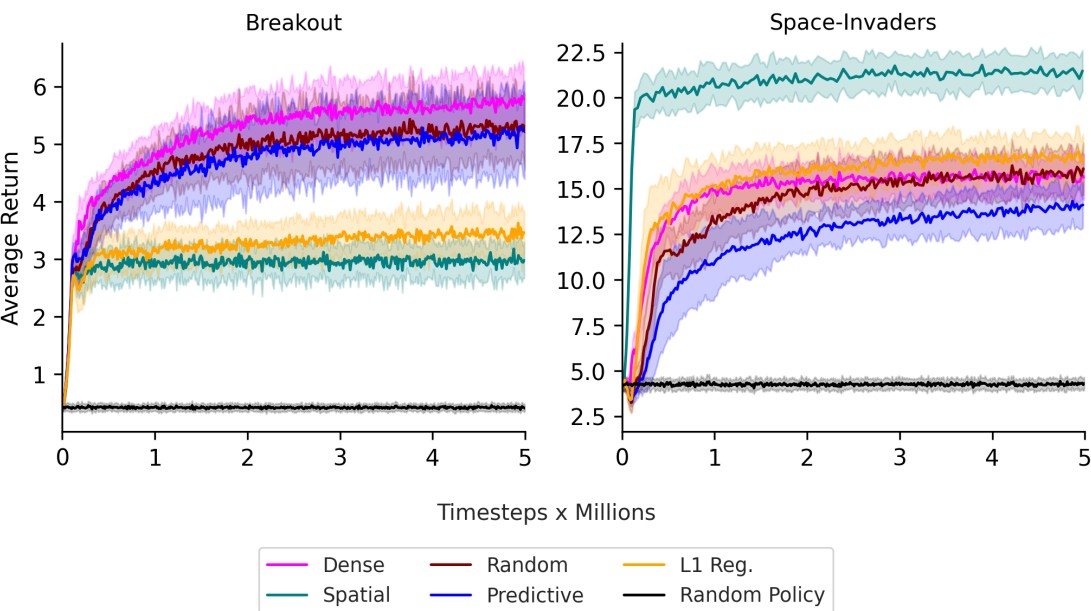

Figure 1: Average return for DQN architectures whose hidden layer is randomly initialized and frozen in each environment: *Breakout* (left) and *Space-Invaders* (right). These results are averaged over 30 runs; the solid lines represent the mean and the shaded regions represent the standard deviation.

In this experiment we initialized the hidden layer weights at random and held them fixed—only allowing the final layer to be learned through back-propagation. We expected that the Spatial architecture would perform best, since it is similar to the sparse structure imposed by convolutional layers which are widely used in domains like MinAtar. Furthermore, as discussed in the previous section, these domains have a distinct spatial structure.

Results from *Breakout* are shown in Figure 1 (left). Our results suggest that Dense achieves the highest performance. We believe this is due to its larger representational capacity (more hidden layer connections). The DQN architectures that achieved the second highest performance were Predictive and Random, although these do not perform significantly different from Dense. Surprisingly, Spatial yields the lowest performance, nowhere near the aforementioned three baselines. The fact that Predictive and Random perform nearly on par with Dense while using significantly less parameters suggests that there are no statistical benefits to using a large fully-connected architecture when the weights are fixed in this domain. In this setting, the simplest sparse architecture (Random) is enough to obtain high performance.

We observe a very different ordering in performance in *Space-Invaders*, as shown in Figure 1 (right). The ordering of the learning curves is almost reversed compared to *Breakout*. Here, Spatial yields the highest average returns—even higher than Dense which has considerably more representational capacity. Similar to the results in *Breakout*, in this domain Random is just as useful as Dense, while Predictive trails behind with the lowest average performance.

In sum, our results suggests that when the hidden layer weights are random values, the choice of sparsity can have a significant impact in the agent's performance. Second, the ordering in performance can also vary significantly across MinAtar domains even with a presumed spatial structure. Lastly, there appear to be no statistical benefits to a fully-connected network compared to sparse architectures with 10 times less hidden layer weights.

## NN Architectures with Learned Hidden Layer

Figure 2: Average return corresponding to DQN architectures whose hidden layer is learned end-to-end in each environment: *Breakout* (left) and *Space-Invaders* (right). These results are averaged over 30 runs; the solid lines represent the mean and the shaded regions represent the standard deviation. Horizontal dashed lines indicate the final average performances when the hidden layer weights are never learned.

### 4.4 Case Study: Learned Hidden Weights and Fixed Topology

Do distinct sparse topologies benefit from learning the weights in the same way? In this case study, we investigate this question by randomly initializing the network and updating the weight magnitudes via backpropagation. We observe that the degree to which performance improves when learning the weights varies across sparse architectures. This gives evidence for our second claim—the performance ranking among sparse structures depends on whether the hidden layer is fixed or learned.

Results for *Breakout* are shown in Figure 2 (left). While Spatial remains the least useful in this domain, we see a change from the learning curves in the previous section: now the average return of Predictive is statistically higher than Random. Learning the weight magnitudes almost doubled the performance of Dense and Predictive (from an average return of nearly 5.5 and 5 to 11 and 10 respectively). Meanwhile, Random and Spatial only improved by approximately 1.3 to 1.4 fold, leaving them behind Predictive and Dense by a significant margin.

Moreover, although the average performance of Predictive is lower than Dense, these are not statistically distinguishable. In other words, there is no statistically significant advantage to using a fully-connected architecture in this domain. Overall, these observations suggest that predictive sparsity indeed provides performance gains when the hidden layer weights are learned end-to-end in RL control, as tested in *Breakout*. Further, this result gives evidence that the utility of a sparse architecture is not just associated to its connections, but also to the combination of connectivity and learned weight magnitudes.

Results from *Space-Invaders* are shown in Figure 2 (right). Here, Spatial incurs the lowest average return. Similar to the results we found in Section 4.3, Predictive under-performs both Random and Dense, now with greater statistical significance. In this environment, learning the hidden layer weights benefited Dense and Random the most: their performance nearly tripled—Dense's performance jumped from 15 to 50, while

Random's increased from 15 to 40 with back-propagation enabled. On the contrary, Spatial's performance barely improved at all, while Predictive's increased but no more than 2 fold. This suggests that in *Space-Invaders* predictive sparsity does not provide performance gains when the hidden layer weights are either fixed or learned end-to-end.

As we expected, in both environments the higher representational capacity of Dense seems to benefit the most from back-propagation. On the other hand, Predictive seems to benefit less compared to Random in *Space-Invaders*. In both environments, even when the hidden layer weights are learned, the spatially-biased sparsity is not useful in RL control. Why, then are convolutional networks—which are spatially biased—so ubiquitous in MinAtar? Is it the fact that convolutional networks have weight sharing that makes their spatially biased kernels so useful? Or, is it their prevalence when applying them to larger environments such as the Arcade Learning Environment (ALE) (Bellemare et al., 2013)? We leave these questions for future work.

So far, all the sparse hidden layer structures have been obtained by a non-end-to-end mechanism (as in the case of Prediction Adapted Networks), or hand-coded like the spatially-biased architecture. However, it remains to be seen whether algorithms that generate network sparsity end-to-end can lead to higher performance gains. In the next section, we investigate how predictive sparsity compares against sparse structures learned end-to-end.

### 4.5 Case Study: Learned Hidden Weights and Topology

In our final case study, we investigate the performance of $L_1$-Reg, an architecture whose topology is learned end-to-end via $L_1$-regularization. Please refer to the appendix for more details on how this type of network sparsity was generated and how we ensured that it has nearly the same number of active connections as the other baselines.

We perform two experiments: we compare the average return of $L_1$-Reg to all other sparse architectures in two scenarios: (1) when the hidden layer weights are randomly initialized and held fixed, and (2) when the latter are learned end-to-end. Due to the greater flexibility that $L_1$-regularization has to mask out weights in a non-uniform fashion throughout the hidden layer, we hypothesize that $L_1$-Reg will yield a higher performance than all the other sparse networks in both environments.

Figure 1 shows the average returns on *Breakout* (left) and *Space-Invaders* (right) for all sparse DQN architectures with fixed hidden layer weights, with $L_1$-Reg shown in yellow. Clearly, our hypothesis is refuted here: on average, $L_1$-Reg performs significantly worse than Random and Predictive in *Breakout* and on par with Random in *Space-Invaders*. Our results suggest that $L_1$-sparsity does not benefit the agent when the hidden layer weights are fixed.

We also investigate how $L_1$-Reg performs when the hidden layer weights are learned, as shown by the yellow learning curves in Figure 2 for *Breakout* (left) and *Space-Invaders* (right). In both environments, we find that $L_1$-Reg performs better than Random and statistically on par with Dense. In this learning regime, there are no benefits to training a fully-connected network, since a network sparsified through $L_1$-regularization can perform just as well.

Why do we observe that $L_1$-Reg performs better than Random, Predictive and Spatial when the hidden layer weights are learned? Recall that in order to generate $L_1$-sparsity, we trained a dense network with a regularized loss function. This means that $L_1$-sparsity is optimized for a network that is learned end-to-end through back-propagation. For this reason, $L_1$-Reg yields high performance when the network is trained, which is precisely what Figure 2 shows. In other words, the learning task that generated $L_1$-sparsity is the same task to approximate the action-values in DQN. On the other hand, the method used to arrive at predictive sparsity—the Prediction Adapted Networks algorithm—is an auxiliary learning mechanism that generates sparse connections in a way that does not directly optimize DQN's off-policy learning objective.

## 5 Conclusion

When we control for network capacity, our empirical results suggest that network sparsity has a significant effect on an agent's performance, and that the ordering in performance among sparse structures depends on whether we fix or learn the hidden layer weights. For instance, in our control domains we observed that relative performance among five sparse networks did not remain consistent when the hidden layer weights were fixed versus learned. Interestingly, in domains with presumed spatial dynamics, spatial sparsity is not the most performant.

Future work may investigate different behaviour polices in the phase where we generate some of our sparse structures, such as predictive and $L_1$-sparsity. Another possibility is to consider more sparse topologies which are expected to be distinct from those considered. One option may include Dynamic Sparse Training techniques (Obando-Ceron et al., 2024; Grooten et al., 2023; Tan et al., 2022; Sokar et al., 2022; Graesser et al., 2022). Since these works are more common in deep-RL, it would be interesting to see if their discovered sparse structures are dominant when the network topology remains fixed while varying the learning process.

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

# A    Experimental Details

## A.1    Trials, Initial Conditions and Environment Dynamics

To evaluate performance, we compute the average return incurred by each DQN agent over 30 independent trials. More specifically, a *trial* refers to a random seed used to (1) randomly initialize the DQN weights at the beginning of learning, (2) sample random actions from an $\epsilon$-greedy policy and (3) sample the environment's initial state. In *Breakout* this amounts to resetting the position of the ball at the start of the game whenever the brick wall is destroyed. On the other hand, in *Space-Invaders*, the environment is fully deterministic and thus does not depend on a random seed.

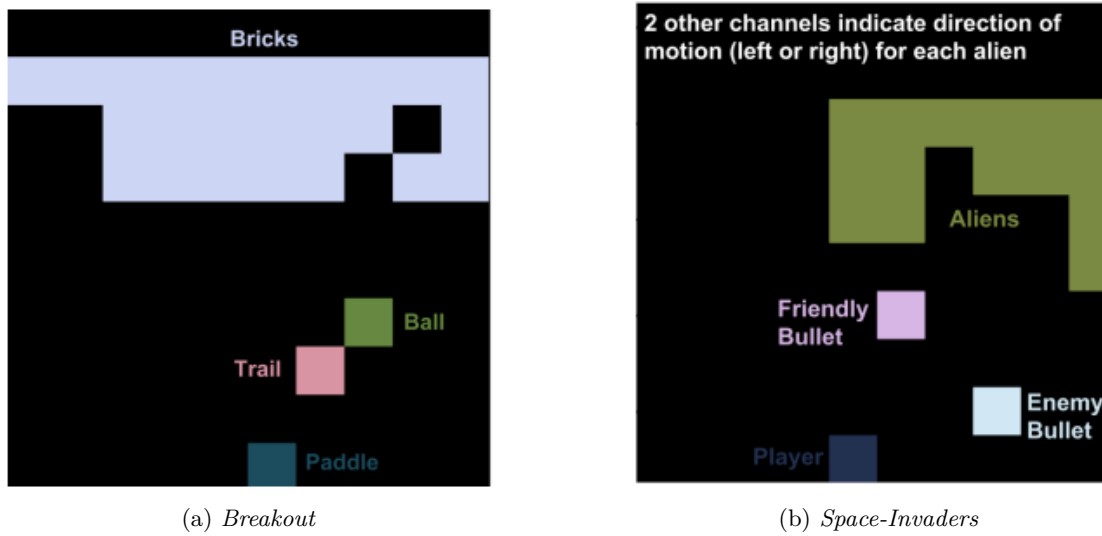

(a) *Breakout*                    (b) *Space-Invaders*

Figure 3: Visualization of the *Breakout* and *Space-Invaders* environments (Young & Tian, 2019).

In *Breakout*, each frame has four $10 \times 10$ input channels corresponding to each of the four objects in the environment: (1) the paddle which moves left or right at the bottom of the frame, (2) the ball which bounces off the paddle, (3) the trail which follows the ball's trajectory one time step in the past and (4) the brick wall. Each time a brick is destroyed, the agent gains a +1 reward; otherwise, the reward is zero. Figure 3a provides a visualization of the *Breakout* environment.

On the other hand, *Space-Invaders* contains six $10 \times 10$ input channels. Four of these correspond to different objects: (1) the cannon representing the player, (2) the aliens representing the enemy, (3) the "friendly bullet" shot by the cannon and (4) enemy bullets shot by the aliens. The extra two channels indicate whether the alien is moving left or right, as described in Figure 3b. The player gets a reward of +1 each time an alien is shot, and that alien is also removed. We note that *Space-Invaders* has non-stationary dynamics: the aliens move with an increased speed when few of them are left, or after a wave of them is fully cleared and a new one appears.

## A.2    Choosing a Percentage of Sparsity

As mentioned in section 4.2, our sparse architectures have hidden layers that are 91% sparse. We chose this percentage to align our work with the setup in the original MinAtar paper (Young & Tian, 2019), where a convolutional layer with a $3 \times 3$ kernel was applied onto the inputs. Thus, we made our hidden layer have 9 active connections per input channel. Since each input channel has 100 entries ($= 10 \times 10$), this amounts to a sparsity level of $(1 - 9/100) \times 100 = 91\%$. This percentage remains at 91% regardless of whether the environment has 4 input channels as in *Breakout*, or 6 input channels as in *Space-Invaders*.

### A.3 Constructing Sparse Architecture Baselines

None of our baselines use convolutional layers. Therefore, in order to feed the 2D observations into our value network, we first flatten the multi-channel inputs to 1D vectors.

Our experiment methodology involves two phases. In phase one, we generate sparse hidden layer topologies and encode them as binary mask matrices. Then in phase two, we impose the binary mask onto DQN's hidden layer weights, as described in section 2.2, and train the agent for 5 million time steps. In the case of Spatial, phase one involves hand-crafting spatially-biased sparse connections. In the case of Random, we assign connections uniformly at random. On the other hand, for Predictive and $L_1$-Reg, generating sparsity is more complex as we explain below.

$L_1$-**Reg:** To generate the $L_1$-sparse baseline, we added $L_1$-regularization to the standard DQN training loss and trained the DQN agent in each environment for 5 million steps. As described in section 4.2, we then took the finalized hidden layer weights whose magnitudes are smaller than the average and zeroed-out their corresponding entries in the binary mask. Figure 10 shows the average hidden layer weight magnitude and the percentage of weights below average in *Breakout*, after sweeping over the $L_1$-regularization coefficient. Figure 11 shows the corresponding plots in *Space-Invaders*. We selected the regularization coefficient whose percentage sparsity came closer to the desired amount shown by the horizontal dashed line: a coefficient of $2.5 \times 10^{-5}$ in *Breakout* and $2 \times 10^{-5}$ in *Space-Invaders*.

**Predictive:** Similarly, to generate the sparse structure of Predictive, we ran Prediction Adapted Networks for 5 million steps with $QV(\lambda)$ as the learning algorithm (Wiering, 2005), where the eligibility trace hyper-parameter $\lambda$ was set to zero. We note that Prediction Adapted Networks is an online and incremental algorithm, hence no replay buffers and no target networks were used. At each time step, Prediction Adapted Networks adapts the connectivity of the value network's hidden layer—the subsets of inputs that make each feature—through an auxiliary learning mechanism. Meanwhile, $QV(\lambda)$ learns the output layer weights. At the end of training, we encoded the final subsets of inputs into the binary mask matrix.

Figures 4, 5 and 6 show examples of hidden layer masks that we generated for the Predictive, Spatial and Random baselines in *Breakout*. Similarly, Figures 7, 8 and 9 show hidden layer masks in *Space-Invaders*.

### A.4 Hidden-layer features and repeated sparsity

In our implementation, each column of the weight matrix generates a single scalar feature. Likewise, each column of the corresponding binary mask encodes a sparse grouping of the inputs. In both environments, the number of distinct groupings is equivalent to the number of inputs. Specifically, in *Breakout*, each grouping is repeated 4 times, thus each set of 4 features were formed from the same grouping. On the other hand, in *Space-Invaders* we repeated each subset of inputs 3 times. These factors were selected in a way that would maintain the number of hidden layer features more of less consistent across both environments. Table 1 lists the dimensionality of the hidden-layer weight matrix in each domain.

## B Experiment Hyper-parameters

Table 1 lists the hyper-parameters we set in both environments. These values define the learning problem setting. Tables 2 and 3 list each architecture's sparsity-generating mechanism, percentage of hidden layer sparsity and step-size used when the hidden layer is fixed versus learned. As mentioned in section 4, we swept over the step-sizes with a grid-search, where we computed the average final return over 5 independent trials; we then selected the step-size that corresponded to the highest average value. Finally, table 4 lists the hyper-parameter values used by the Prediction Adapted Neighbors and $QV(\lambda)$ algorithms to generate the sparse structure of Predictive.

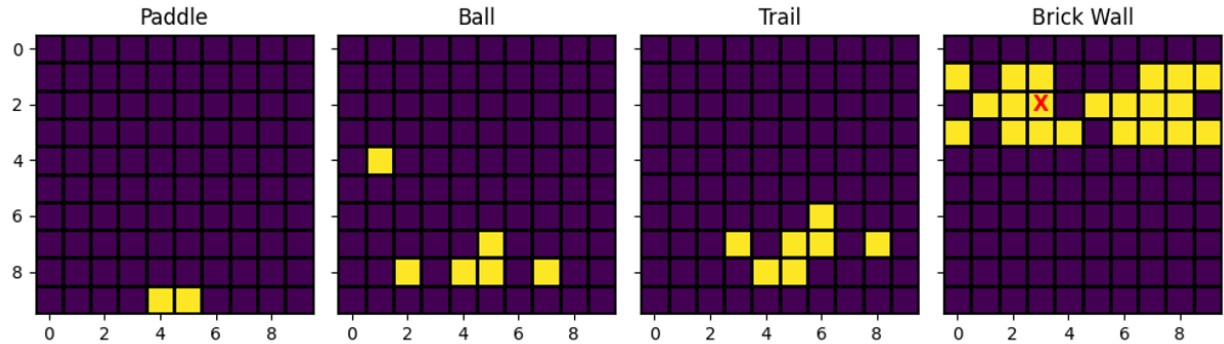

Figure 4: One of the predictive masks in *Breakout*. The inputs in yellow are those that are "on" in the mask. Prediction Adapted Neighborhoods found this subset to help predict the next values of the entry marked by the red "X".

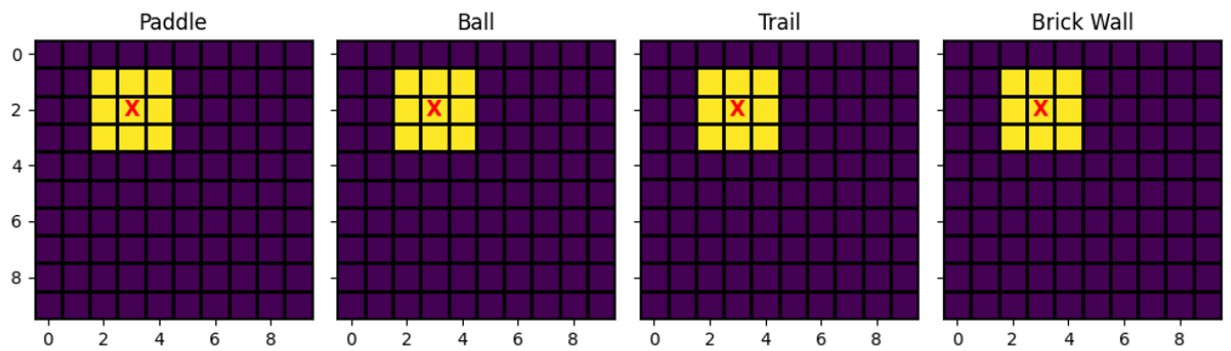

Figure 5: One of the spatial neighborhoods in *Breakout*. The inputs that are "on" are shown in yellow—these are located closest to entries marked by the red "X".

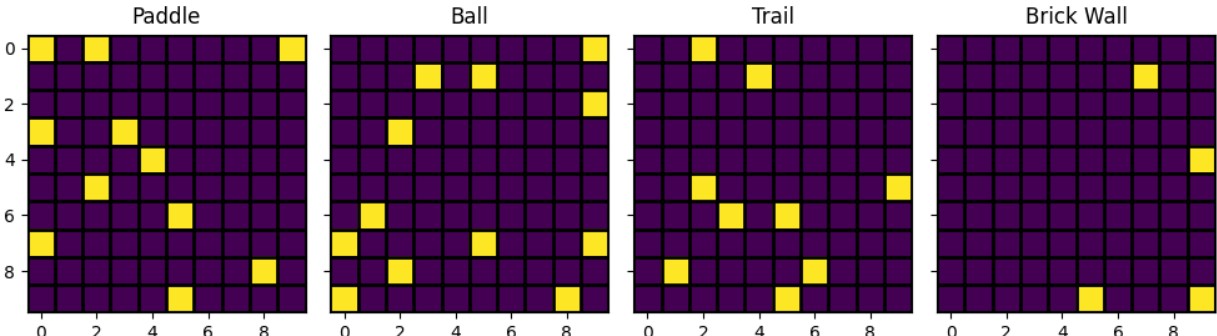

Figure 6: One of the random neighborhoods in *Breakout*.

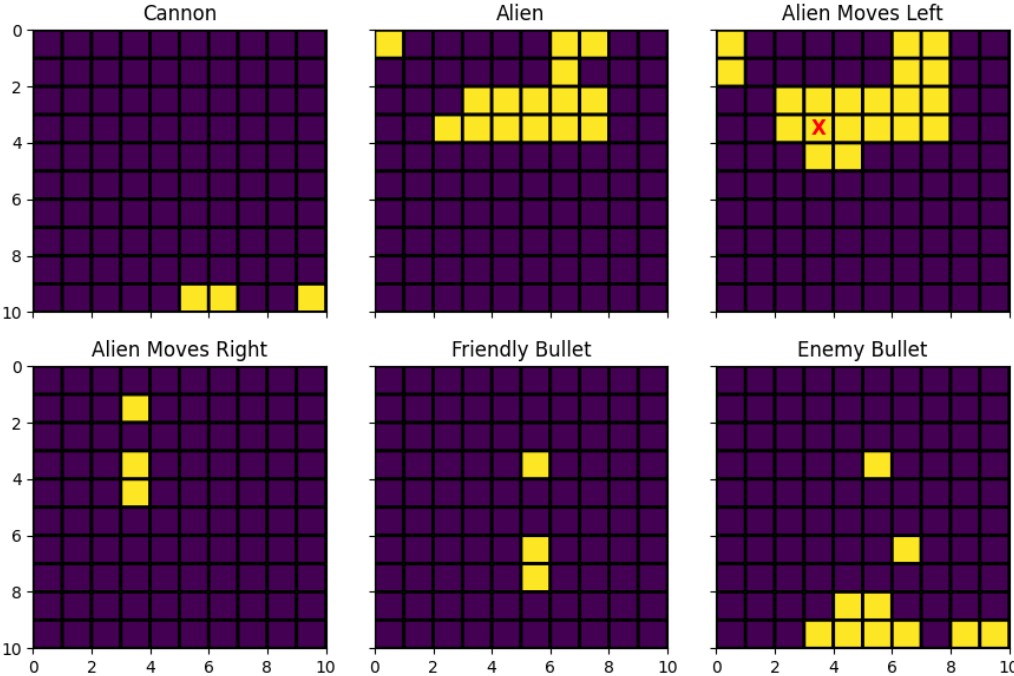

Figure 7: One of the predictive masks in *Space-Invaders*. The inputs in yellow are those that are "on" in the mask. Prediction Adapted Neighborhoods found this subset to help predict the next values of the entry marked by the red "X".

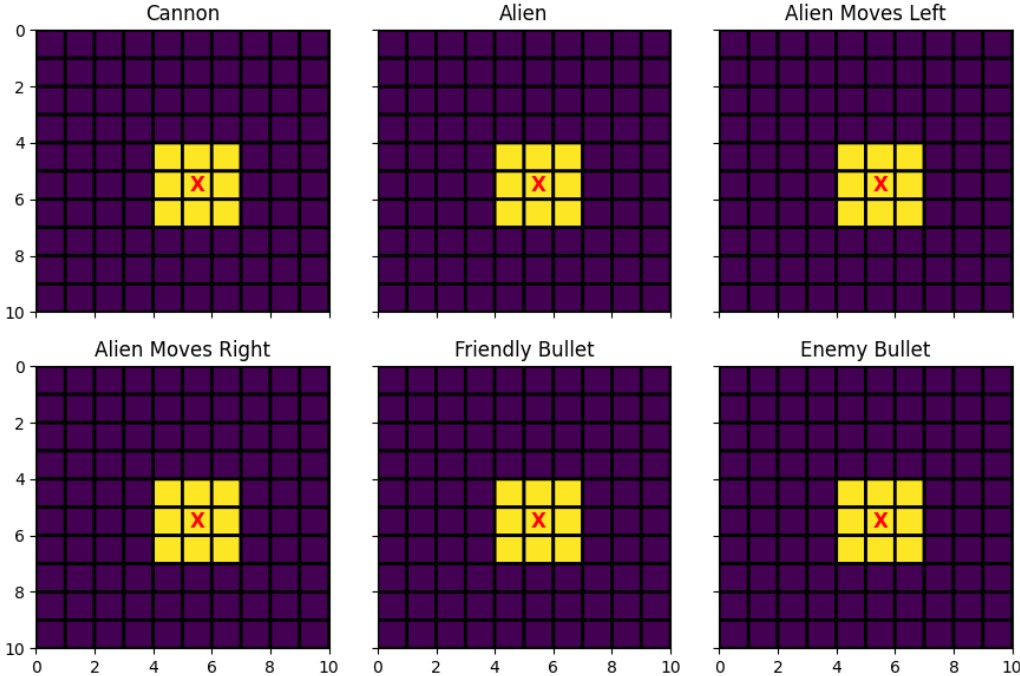

Figure 8: One of the spatially-biased masks in *Space-Invaders*. The inputs that are "on" are shown in yellow—these are located closest to entries marked by the red "X".

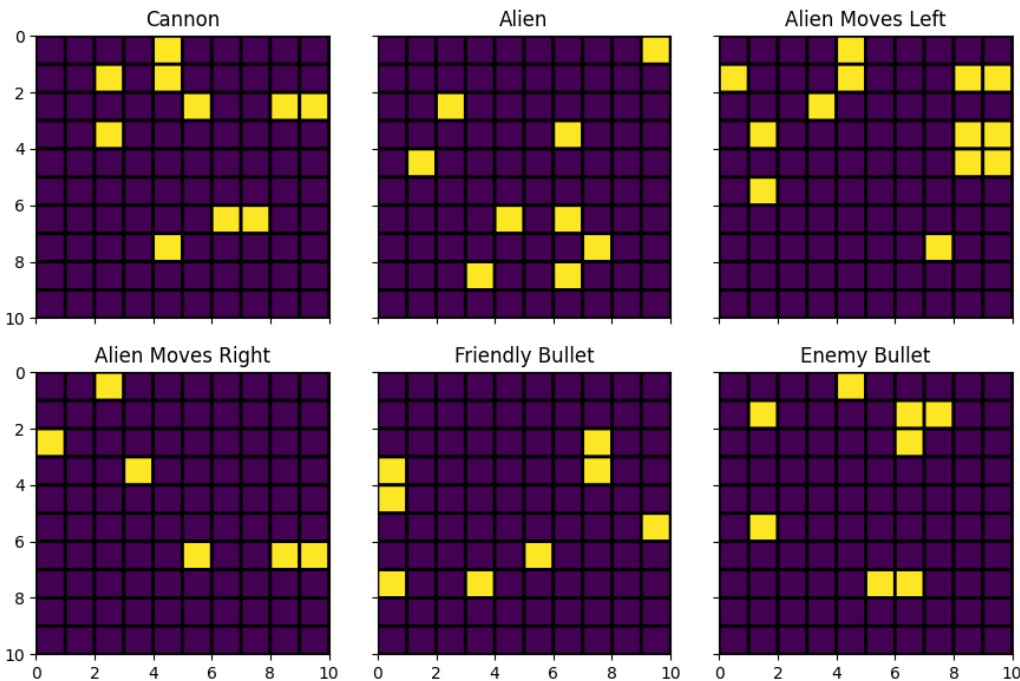

Figure 9: One of the random binary masks in *Space-Invaders*.

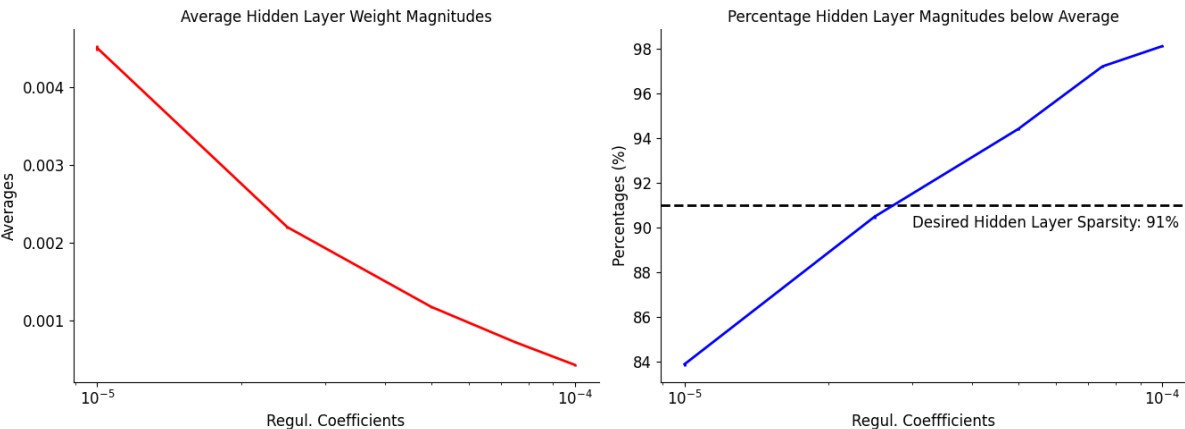

Figure 10: Average magnitudes (left) and average percentages (right) of hidden layer weights that fall below average, with respect to regularization coefficients in *Breakout*. Both statistics were computed over 30 independent trials. Error bars are shown as vertical line segments.

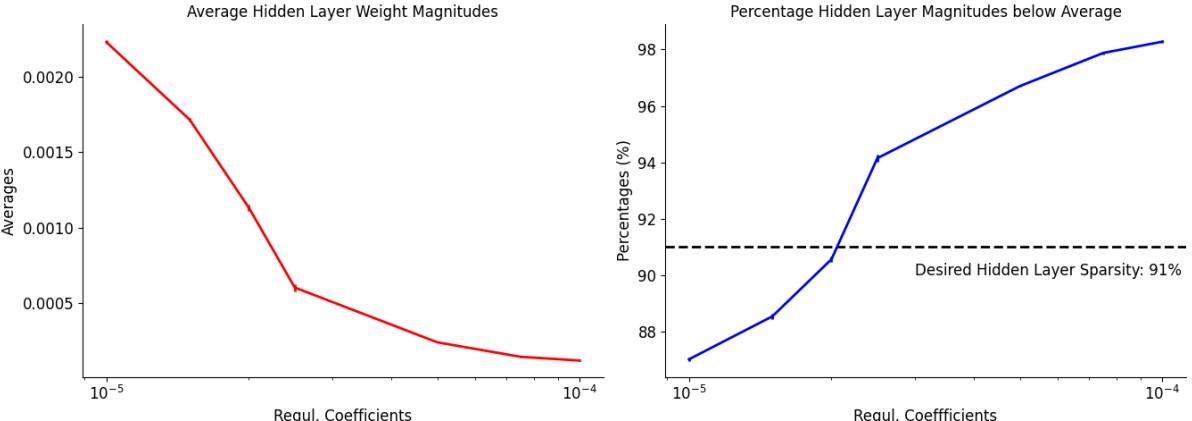

Figure 11: Average magnitudes (left) and average percentages (right) of hidden layer weights that fall below average, with respect to regularization coefficients in *Space Invaders*. Both were computed over 30 independent trials. Error bars are shown as vertical line segments.

|  | Environments | |
|---|---|---|
| **Hyper-parameters** | *Breakout* | *Space-Invaders* |
| Discount Factor $\gamma$ | 0.99 | 0.99 |
| Exploration parameter $\epsilon$ | 0.1 | 0.1 |
| HL matrix dimension | $400 \times 1600$ | $600 \times 1800$ |
| OL matrix dimension | $1600 \times 3$ | $1800 \times 4$ |
| HL bias units | Yes | Yes |
| OL bias units | Yes | Yes |
| Non-linear Activation Function | ReLU | ReLU |
| Number of Timesteps | 5M | 5M |

Table 1: Hyper-parameters specifying the DQN architecture, discount factor and exploration parameter for each environment.

| | | Hyperparameters in Breakout | | |
|---|---|---|---|---|
| **Architectures** | Sparsif. Mechanism | Step-Size HL is frozen | Step-Size HL is learned | % HL Sparsity |
| **Predictive** | Prediction Adapted Networks | 0.1 | 0.0001 | 91 |
| **Random** | Uniformly at random | 0.1 | 0.0001 | 91 |
| **Spatial** | Hand-crafted | 0.1 | 0.0001 | 91 |
| $L_1$-**reg** | $L_1$-regularization | 0.1 | 0.0001 | 90.88 |

Table 2: Hyper-parameters used in each sparse architecture in *Breakout*. The table lists the sparsification mechanism, final step-sizes used when the hidden-layer is frozen vs. learned and the final percentage sparsity in the hidden-layer.

| | | Hyperparameters in Space-Invaders | | |
|---|---|---|---|---|
| **Architectures** | Sparsif. Mechanism | Step-Size HL is frozen | Step-Size HL is learned | % HL Sparsity |
| **Predictive** | Prediction Adapted Networks | 0.0001 | 0.0001 | 91 |
| **Random** | Uniformly at random | 0.0001 | 0.0001 | 91 |
| **Spatial** | Hand-crafted | 0.0001 | 1e-05 | 91 |
| $L_1$-**reg** | $L_1$-regularization | 0.0001 | 1e-05 | 89.98 |

Table 3: Hyper-parameters used in each sparse architecture in *Space-Invaders*. The table lists the sparsification mechanism, final step-sizes used when the hidden-layer is frozen vs. learned and the final percentage sparsity in the hidden-layer in each environment respectively.

| | Environments | |
|---|---|---|
| **Hyper-parameters** | *Breakout* | *Space-Invaders* |
| **Number of neighbors** $k$ | 9 | 9 |
| **Number of general value functions (GVF)** $m$ | 400 | 600 |
| **GVF step-size** $\bar{\alpha}$ | 3e-06 | 3e-06 |
| **GVF eligibility-trace parameter** $\bar{\lambda}$ | 0 | 0 |
| **GVF discount factor** $\bar{\gamma}$ | 0.99 | 0.99 |
| **Number of pre-activations per neighborhood** $n$ | 16 | 16 |
| **Pre-activation bias unit** | 0 | 0 |
| **Frequency of neighborhood updates (in time steps)** | 1000 | 1000 |
| **QV(0) step-size** $\alpha$ | 5e-06 | 5e-06 |
| **QV(0) discount factor** $\gamma$ | 0.99 | 0.99 |
| **QV(0) exploration parameter** $\epsilon$ | 0.1 | 0.1 |

Table 4: Prediction Adapted Networks and QV(0) hyper-parameters used to generate the sparse structure of Predictive. We show three QV(0) hyper-parameters in the bottom rows, the remaining rows pertain to Prediction Adapted Networks.

