# OpenReview forum: "On the Interplay Between Sparsity and Training in Deep Reinforcement Learning"
_TMLR — Rejected by TMLR_

### Review · Reviewer_G7Tm · 2025-02-11

**Summary Of Contributions:**

The paper investigates the impact of different sparse architectures on deep reinforcement learning, specifically in image-based domains. The authors explore spatially-biased and fully-connected architectures, among others, of equal capacity. They study how the sparse structures influence learning in two tasks in MinAtar using DQN.

**Audience:**

No

**Broader Impact Concerns:**

There is no border impact concern

**Claims And Evidence:**

No

**Requested Changes:**

Please see the discussion mentioned in "Strength and weakness" section.

**Strengths And Weaknesses:**

* The authors conducted two experiments to compare the performance of sparse-architecture: (1) with fixed and (2) learnable hidden layers during training. However, there are several issues with their experiments and conclusions that need clarification:
    * Fixed Hidden Layer Experiment: Since this is not practically used, it is unclear why the authors chose to fix the hidden layer during training. The experimental design is not clear to me and the results do not seem to provide clear insights, they are contradictory. Without understanding when and why one approach should work, the experiment does not lead to any solid conclusions.
    * Learnable Hidden Layer Experiment: It is obvious that a learnable hidden layer should improve performance, similar to any other neural network training.
* In my opinion the paper discusses three types of sparse architecture: predictive, spatially biased, and dense networks sparsified using L1 and random sparsity. However, L1 and random sparsity are not considered state-of-the-art methods in reinforcement learning. The paper references Rigl [1], DST[2] and does not include them in their experiment. It also ignores more recent advancements in sparse-RL, such as RLX2 [3], DPAD [4], and [5]. The authors should discuss how their findings relate to these works and how can be generalized to these methods.
* The paper discusses the benefits of different "sparse architectures" but does not clearly define what “sparse architecture” means. This term is not commonly used in the literature, so the authors should properly define this term.
* The paper focuses on an image-based domain but does not consider using convolutional neural networks (CNNs), which are used for image tasks in RL for the best performance. It requires further explanation as to why CNNs were not included.
* The authors use a sparsity level of 91%, but they do not explain why this specific choice was made. How does the change in sparsity affect their findings?
* There are other hyperparameters that impact the performance of sparse architectures, such as the learning rate and the size of the hidden dimension. Do the authors' findings hold the same impact when these hyperparameters are varied?
* The experiments in the paper are limited to only two tasks. How can the conclusions drawn from these experiments be applied to a broader range of scenarios?

---
Reference:
1. The State of Sparse Training in Deep Reinforcement Learning. https://proceedings.mlr.press/v162/graesser22a/graesser22a.pdf
2. Dynamic Sparse Training for Deep Reinforcement Learning. https://arxiv.org/abs/2106.04217
3. RLx2: Training a Sparse Deep Reinforcement Learning Model from Scratch. https://arxiv.org/abs/2205.15043
4. In value-based deep reinforcement learning, a pruned network is a good network. https://arxiv.org/pdf/2402.12479
5. Efficient Reinforcement Learning by Discovering Neural Pathways. https://openreview.net/forum?id=WEoOreP0n5

---

> ### Author Response · Authors · 2025-03-04
> **Response to the reviewer's first comment**
>
> Thank you for your comments. Please find our responses below. We will also update a revised version of the paper with the modified sections in red font.
>
> > Fixed Hidden Layer Experiment: Since this is not practically used, it is unclear why the authors chose to fix the hidden layer during training. The experimental design is not clear to me and the results do not seem to provide clear insights, they are contradictory.
>
> Thank you for your feedback. Our experiment design aimed to investigate the relative ordering in performance of various sparse structures with respect to learning or fixing the hidden layer weights. The question we want to answer is: does the ordering in performances remain the same when the weights are learned? In other words, does the best performing sparse architecture with fixed hidden layer weights remain dominant when those weights are learned? The reason why we fixed the hidden layer weights was not because doing so is practically used. Rather, we included this experiment because we wanted to decouple the performance gains due to sparse structure from those due to backpropagation; fixing the weights is a means to control for the latter.
>
> Moreover, there exist RL algorithms that maintain the hidden layer weights fixed to random values, such as PANs and Generate-and-Test (Martin and Modayil, 2021; Mahmood and Sutton, 2013). In Generate-and-Test, hidden layer weights are randomly initialized to $\pm 1$. These weights are never learned through backpropagation; rather, if their associated output features have low utility, the weights are dropped and replaced with newly sampled $\pm 1$ values. Although these two algorithms are not practically used in applications, this does not mean that they should be discarded. The same is true for our fixed hidden-layer case-study.
>
> > Without understanding when and why one approach should work, the experiment does not lead to any solid conclusions.
>
> Thank you. We were unclear whether “approach” in your comment refers to a sparse structure (Spatial, Random, Predictive, L1) or to the learning process (fixing versus learning the weights).
>
> Assuming that you meant the former, our paper provided a hypothesis as to why we believed that Spatial should outperform the other sparse structures -- this is because our image-based domains have an apparent spatial dependence. As we mentioned in Section 4.3: “We expected that the Spatial architecture would perform best, since it is similar to the sparse structure imposed by convolutional layers which are widely used in domains like MinAtar.”
>
> To clarify, when we claimed that our domains have an apparent spatial dependence, we meant that the interactions between objects in the domains tend to be spatially local. For instance, in Breakout the trail is always found one spatial unit diagonally behind the ball. Moreover, when the ball hits a brick on the wall, all adjacent bricks will likely be taken down as well. In Space-Invaders, the aliens remain stuck to one another and move left or right in a united front; enemy bullets move downwards from the aliens, while friendly bullets move upwards from the cannon’s location.
>
> Assuming that “approach” refers to the learning process: a network architecture with learned weights will perform better compared to having weights fixed to random values. The reason why we varied whether we learn the hidden layer weights was explained in our response to the first comment.
>
> **References:**
>
> John D Martin and Joseph Modayil. Adapting the function approximation architecture in online reinforce-
> ment learning. arXiv preprint arXiv:2106.09776, 2021.
>
> Mahmood, A. R., and Sutton, R. S. (2013, June). Representation Search through Generate and Test. In AAAI Workshop: Learning Rich Representations from Low-Level Sensors

---

> > ### Author Response · Authors · 2025-03-04
> > **Response to reviewer's second, third and fourth comments**
> >
> > Thank you for the feedback, we attached our responses below.
> >
> > > In my opinion the paper discusses three types of sparse architecture: predictive, spatially biased, and dense networks sparsified using L1 and random sparsity. However, L1 and random sparsity are not considered state-of-the-art methods in reinforcement learning. The paper references Rigl [1], DST[2] and does not include them in their experiment. It also ignores more recent advancements in sparse-RL, such as RLX2 [3], DPAD [4], and [5]. The authors should discuss how their findings relate to these works and how can be generalized to these methods.
> >
> > Thank you for pointing this out and for providing these references; we included them references in a revised version of the document.
> >
> > We do not claim that random sparsity and L1 are state-of-the-art methods in RL. Our baselines were chosen as methodological devices for comparing different sparse structures. Our chosen baselines have (1) distinct topologies and (2) were generated from different techniques. We chose Random to represent unstructured sparsities and L1 to represent architectures whose sparsity is an artifact of the learning process.
> >
> > Indeed, it would be good to include RigL and DST techniques, but we leave this for future work. Our goal was to provide evidence for how the ordering in performance among sparse architectures depends on the context — the input data distribution and whether we fix or learn the hidden layer weights (i.e., the learning process itself). We found that architectures that work best when the hidden layer is fixed do not necessarily remain dominant when the hidden layer is learned; knowing this is important to understand what makes sparsity beneficial.
> >
> > > The paper discusses the benefits of different "sparse architectures" but does not clearly define what “sparse architecture” means. This term is not commonly used in the literature, so the authors should properly define this term.
> >
> > Thank you for bringing this to our attention. We use the term sparse architecture interchangeably with sparse network, meaning a neural network with a large number of zero-valued parameters. We clarified this in the revised version of the paper.
> >
> > > The paper focuses on an image-based domain but does not consider using convolutional neural networks (CNNs), which are used for image tasks in RL for the best performance. It requires further explanation as to why CNNs were not included.
> >
> > Thank you. CNNs were not included because they introduce sources of variation that we cannot control. As mentioned in the paper, “We also consider a spatially-biased baseline, Spatial. This architecture is similar to a convolutional layer in how it forms receptive fields with nearby pixels. However, to control for the effects of sparse structure and control for the number of learnable parameters, Spatial does not impose weight sharing; each kernel contains its own learnable weights.”
> >
> > It is true that CNNs are typically used for image tasks. However, they impose weight-sharing and are equivariant to translation (LeCun et al., 2015). Our experiments focus on the effects or sparse structure and the learning process on the relative ordering of performance, not other extraneous factors. For instance, if the ranking of a convolutional net became higher or lower compared to the other networks when the weights are learned, we would not know which of its properties (spatially-biased sparse structure, weight sharing or equivariance) accounts for this.
> >
> > We clarified this in the revised version of the document.
> >
> > \
> > **References:**
> >
> > Yann LeCun, Yoshua Bengio, and Geoffrey Hinton. Deep learning. Nature, 521(7553):436–444, 2015.

---

> > > ### Author Response · Authors · 2025-03-04
> > > **Response to reviewer's fifth, sixth and seventh comments**
> > >
> > > Thank you for your feedback. Our responses are enclosed below.
> > >
> > > > The authors use a sparsity level of 91%, but they do not explain why this specific choice was made. How does the change in sparsity affect their findings?
> > >
> > > Thank you very much for pointing this out to us. We chose this percentage to align our work with the setup in the original MinAtar publication (Young and Tian, 2019), where a convolutional layer with a $3 \times 3$ kernel was applied to the inputs. Thus, we made our hidden layer have 9 active connections per input channel. Since each input channel has 100 entries ($= 10 \times 10$), this amounts to a sparsity level of $(1 - 9/100) \times 100 = 91\%$. Note, this percentage remains at 91\% regardless of whether the environment has 4 input channels as in Breakout, or 6 input channels as in Space-Invaders.
> > >
> > > We added a clarification in the revised version of the document (Appendix, section A.2).
> > >
> > > > There are other hyperparameters that impact the performance of sparse architectures, such as the learning rate and the size of the hidden dimension. Do the authors' findings hold the same impact when these hyperparameters are varied?
> > >
> > > Thank you for raising this question. We swept over the step-size with a grid-search, where we computed the average final return and standard deviation over 5 independent trials and selected the step-size that corresponded to the highest value. We do not have the step-size ablation data, but the code that executed the grid-search will be made available.
> > >
> > > We did not sweep over the number of hidden layer features because our goal was not to investigate the effects or sparsity across different hidden layer feature counts. Rather, we wanted to limit the scope of our analysis to (1) different sparse hidden layer structures and (2) the learning process. All sparse architectures have the same number of hidden layer weights and hidden layer features in each environment. Furthermore, our choice of hidden layer feature counts were enough to match the performance shown in the original MinAtar paper when the hidden layer was learned.
> > >
> > > > The experiments in the paper are limited to only two tasks. How can the conclusions drawn from these experiments be applied to a broader range of scenarios?
> > >
> > > Indeed, there could be some aspect of larger domains that invalidates our results. Other than scale, it is unclear to us what this aspect might be. The reason why we chose two MinAtar domains was to provide a stand-in for environments with a presumed spatial structure. Furthermore, MinAtar is a testbed where we can investigate our hypotheses without needing large compute resources.
> > >
> > > \
> > > **References:**\
> > > Kenny Young and Tian Tian. MinAtar: An Atari-inspired testbed for thorough and reproducible reinforce- ment learning experiments. arXiv preprint arXiv:1903.03176, 2019.

---

### Review · Reviewer_Aebq · 2025-02-13

**Summary Of Contributions:**

The paper deals with the influence of sparse structures in the neural network to represent the Q-function in the DQN algorithm, an approach of model free online reinforcement learning.
It confirms the already known observation that sparsity has an influence.

**Audience:**

Yes

**Claims And Evidence:**

Yes

**Requested Changes:**

If I assume that the paper does not intend to present a new method or a new theory, but to make a scientific contribution through an empirical study, then this study must be much more extensive.

* More environments could be studied,
* more algorithms from the class of model-free online RL could be studied,
* offline RL methods could be studied,
* methods of model-based RL could be studied.
* Sparse networks could be studied not only to represent the Q-function, but also
* to represent the policy or
* to represent the transition model.

I am not saying that all these aspects need to be investigated, I am just suggesting directions in which the study could be extended. And in my opinion, an extension is absolutely necessary in order to present sufficient new insights for the readers of the TMLR.

Further comments:

* When discussing the preliminary work on “Sparse Networks in Supervised Learning”, a more complete listing would increase the quality, e.g. by citing early work, such as [1] and possibly also review articles, such as [2]

* In the paragraph “Connection between Representations and the environment” the capitalization is not consistent.

* Figures 1 and 2 do not explain what the shaded area is.

* In my opinion, such important information as “we compute the average ... over 30 independent trials” should not only appear in the Appendix, but in the main text.

[1] Zimmermann et al., Forecasting with Recurrent Neural Networks: 12 Tricks. In: Neural Networks: Tricks of the Trade, 2012\
[2] Hoefler et al., Sparsity in Deep Learning: Pruning and growth for efficient inference and training in neural networks, JMLR 2021

**Strengths And Weaknesses:**

**Strengths**
* The paper is well written
* The learning experiments were repeated pleasantly often (30 times)

**Weaknesses**
* The knowledge gained is very limited. No new method and no new theory is presented, only an empirical study. In my opinion, the study would have to be much more extensive to provide sufficient added value.

---

> ### Author Response · Authors · 2025-03-04
> **Response to the reviewer's comments**
>
> Thank you very much for your feedback. Please find our responses below. We will attach a revised version of the document.
>
> > When discussing the preliminary work on “Sparse Networks in Supervised Learning”, a more complete listing would increase the quality, e.g. by citing early work, such as [1] and possibly also review articles, such as [2].
>
> Thank you. We included these two citations in the Related Work section of the revised document.
>
> > In the paragraph “Connection between Representations and the environment” the capitalization is not consistent.
>
> Thank you for pointing this out. We fixed this in the revised version of the document.
>
> > Figures 1 and 2 do not explain what the shaded area is.
>
> Thank you, this is now included.
>
> > In my opinion, such important information as “we compute the average ... over 30 independent trials” should not only appear in the Appendix, but in the main text.
>
> Thank you. This was included already, above section 4.1.

---

### Review · Reviewer_tKbn · 2025-02-17

**Summary Of Contributions:**

This work examines the impact of different sparse architectures on deep reinforcement learning, which is focused on image-based domains where "spatially-biased and fully connected structures are common". The study compares different architectures with equal capacity.

**Audience:**

Yes

**Claims And Evidence:**

No

**Requested Changes:**

- Empirical results that can actually be used to draw meaningful conclusions
- Fix a few issues in the formalization
- Can the source code be shared?

**Strengths And Weaknesses:**

The work mentions that it focuses on "image-based" domains, however it only uses MinAtar which limits the scope of the experimental conclusion. Indeed, MinAtar is a very simplified "image-based" domain that removes a large part of the difficulty of learning based on images. I would suggest also having at least another domain of environments.

In Figure 1, given the very limited set of experiments (2 MinAtar environments), it seems complicated to draw any conclusion. This is clearly stated by the authors "We observe a very different ordering in performance in Space-Invaders, as shown in Figure 1 (right). The ordering of the learning curves is almost reversed compared to Breakout.".

It would be worth getting more empirical evidence before this paper can be accepted. In addition, there are a few elements that needs to be fixed (see a few of them in the additional comments below).

Additional comments:
- In the abstract  it is written "image-based domains where spatially-biased and fully connected structures are common" but it does not seem to me that fully connected structures (e.g. without CNN) are often used when working directly from pixels.
- In the problem setting, it is written that "the agent chooses an action $a_t$ based on its current observation $o_t$". But this does not seem coherent with (1) the fact that it not only uses *current* observation later in the paragraph and (2) the definition of the histories where the agent is supposed to decide on the actions based on n pairs of (actions, observations): for instance at t=0, it seems the agent needs to choose an action first without having access to any observation.
- In 2.1, it's unclear why $\hat f$ is introduced. The hypothesis for getting rid of the partially observability does not seem clearly stated.
- In Figure 1, the average return seems very low (for instance max 5 for the game breakout ?).

---

> ### Author Response · Authors · 2025-03-04
> **Response to the reviewer's comments**
>
> Thank you for all the comments and feedback. Please find our responses below.
>
> > The work mentions that it focuses on "image-based" domains, however it only uses MinAtar which limits the scope of the experimental conclusion. Indeed, MinAtar is a very simplified "image-based" domain that removes a large part of the difficulty of learning based on images. I would suggest also having at least another domain of environments.
>
> Thank you for the suggestion. Other than scale, it is unclear to us what other aspect of larger domains would invalidate our results. The reason why we chose MinAtar was to provide a stand-in for domains with a presumed spatial structure. MinAtar is a testbed where we can investigate our hypotheses without needing large compute resources.
>
> > In Figure 1, given the very limited set of experiments (2 MinAtar environments), it seems complicated to draw any conclusion. This is clearly stated by the authors "We observe a very different ordering in performance in Space-Invaders, as shown in Figure 1 (right). The ordering of the learning curves is almost reversed compared to Breakout.". It would be worth getting more empirical evidence before this paper can be accepted.
>
> Thank you for this feedback. Our goal was not to show that a single sparse structure outperforms the others in both domains. Our experiment design is meant to help us understand how the learning process (learning versus fixing the hidden layer weights) affects the relative ordering in performance among sparse structures in each domain. We concluded that the best performing sparse architecture with fixed weights does not necessarily remain dominant when the weights are learned.
>
> The fact that there was a different ordering in performance in Space-Invaders versus Breakout suggests that the benefits of sparsity also depend on the input distribution. As we point out in our introduction, there is a strong connection between sparse structure and the interdependencies among inputs.
>
> **Addressing additional comments:**
> > In the abstract it is written "image-based domains where spatially-biased and fully connected structures are common" but it does not seem to me that fully connected structures (e.g. without CNN) are often used when working directly from pixels.
>
> Thank you, your comment is correct here. We will revise the paper to state “image-based domains where spatially-biased structures are common, such those provided by convolutional networks”.
>
> > In the problem setting, it is written that "the agent chooses an action a_t based on its current observation o_t". But this does not seem coherent with (1) the fact that it not only uses current observation later in the paragraph and (2) the definition of the histories where the agent is supposed to decide on the actions based on n pairs of (actions, observations): for instance at t=0, it seems the agent needs to choose an action first without having access to any observation.
>
> Thank you for pointing this out. A couple clarifications.
> 1. We have revised the manuscript to say "the agent chooses an action $a_t$ after receiving the observation $o_t$."
> 2. Equation 1 is written so it applies to any history and action.
>
> > In 2.1, it's unclear why $\hat{f}$ is introduced. The hypothesis for getting rid of the partially observability does not seem clearly stated.
>
> Thank you for this question. Partial observability is not assumed for this work. We introduce $\hat{f}$ as the state-update function; this mathematically represents the body of the DQN architecture. The agent's internal state can thus be viewed as parameters of a neural network. The input to the network is the observation (i.e. an image), and the outputs are the values of each action. We will clarify this in our revision.
>
> > In Figure 1, the average return seems very low (for instance max 5 for the game breakout ?).
>
> In Figure 1, we do not train the DQN hidden layer, which accounts for the lower performance. On the other hand, Figure 2 (left) shows the learning curves in Breakout when the hidden layer is learned end-to-end. In this figure, we see that the final performance of Predictive, Dense and L1 closely match the one reported in the original MinAtar paper (Young and Tian, 2019).
>
> > Can the source code be shared?
>
> Yes, the code will be shared as a zipped folder to maintain anonymity.
>
> We will update a revised version with any modifications in red font.

---

### Decision · Action_Editor_3Yjr · 2025-04-06

**Recommendation:** Reject

**Comment:**

This empirical paper investigates the effect of NN sparsity in RL.
It specifically studies two aspects:
(1) whether the choice of sparse architecture (random, following the spatial regularity of the input, induced by L1 regularization, etc.) has any effect on the performance.
(2) whether using a fixed vs. learnable representation layer (hidden layer) has any effect on the performance.

They perform experiments on two MinAtar environments and observe that (briefly)
(1) yes, different sparsity structures have different performances, and their relative ranking is different on each of two environments.
(2) yes, there is a difference in ranking between fixed vs. learnable representation.

The final recommendation of the reviewers are as follows:
**Reject, Leaning Reject, and Leaning Accept**.

Given that two of three reviewers are on the negative side, it appeared to me that the paper could not convince some RL experts about its merit in its current form (the positive reviewer is also critical and not a champion of the paper). To ensure that the reviewers' responses weren't due to random variations, I read the paper myself, before carefully reading the reviews. I agree with the reviewers on several points to believe that the paper has room for becoming much stronger after some revisions. These modifications are beyond the level of a minor revision, hence **I recommend rejection with the encouragement to revise and resubmit**.


Let me explain some of the positive and negative aspects of the paper, based on reviewers and my own reading:

On the positive side, I agree with some of reviewers that this paper performs an interesting study. It might not be groundbreaking or significantly novel, but the paper asks a valid research question and tries to answer it.

I also share the positive comment by one of the reviewers that the number of runs in experiments is 30, which is a good methodological practice not common among most deep RL papers.


On the other hand, there are some limitations:

* A common raised concern among the reviewers in their final recommendation, as well as in their original review, was the limitation of the experiments. They believe that this limited the insights one could gain from the paper. This was even shared by the more positive reviewer.


The limitation of experiments are in multiple aspects. For example:
1) Only including MinAtar, and not including other image-based domains.
2) Only two MinAtar environments
3) Not comparing with a larger variety of state of the art sparsity inducing methods.
4) Focusing on only one RL algorithm.

These limitations vary in significance:
For example, regarding item (1), the authors replied that they only included MinAtar because it has some spatial structure common in image-based tasks and is a less of a computational burden (and this is presumably the reason why they could conducts experiments for 30 runs). This rationale is understandable, though still a limitation -- perhaps not a major one though. I believe the paper would be much better and its conclusions broader if the type of environments are vastly different.


I find item (2) (only two environments) less justified.
To be precise, having two environments is enough to disprove a claim such as "the ranking of sparsity methods is always the same across environments" (one variation is enough to falsify the claim), but this insight is rather limited. Does it usually happen? Or does it occasionally happen? The information gained from "different ranking in 8 out of 8 environment" is different from "different ranking in 1 out of 8 environments". The current experiments at best can show the evidence of the latter.

Regarding 3) Comparing with a wider variety of sparse NN methods, especially the state of the art, may not be absolutely necessary, but certainly makes the paper stronger and more relevant to the current literature.

Regarding 4) This is optional, but including more than one algorithm (perhaps a policy search-based one) is a very good way to make the paper stronger and its conclusions broader.

* Not including CNN is a limitation of the paper.
I see that there is less control over the CNN due to the weight sharing, as mentioned by the authors, but CNN is such a common architecture in image-based methods that not including it, perhaps even in one comparison, shall raise questions. I suggest that the authors include such a comparison in their next revision.


I have two other concerns, which are not raised by reviewers:

* There are some mentions of **statistical significance** of the difference between methods. For instance, it is said that "although the average performance of Predictive is lower than Dense, these are not statistically distinguishable. In other words, there is no statistically significant advantage to using a fully-connected architecture in this domain".

Where does this claim come from? Does it come from the observed overlap between the shaded area, which are one standard deviation around the mean?
Such an overlap is not enough for claiming no statistical significance. Just imagine two random variables X coming N(0,1) and N(0.5,1). If we draw standard deviations around 0 and 0.5, they will overlap, but their means are certainly different.


* *The architecture sparsity of a NN and the sparsity compatible with the local spatial structure of the input data are two different things*. They coincide when there is only one hidden layer, but if the NN is deeper, they may not directly relate to each other.
Since this work only experiments with NN with one hidden layer, this is not a major issue, but if the claims are about the architectural sparsity no matter the depth, this distinction should be clarified better.

**Audience:**

Yes, this paper could be very relevant to RL researchers.

**Claims And Evidence:**

No, it is rather limited. See the comments.

**Resubmission Of Major Revision:**

The authors may consider submitting a major revision at a later time.